# High-dimensional analysis of intestinal immune cells during helminth infection

Laura Ferrer-Font, Palak Mehta, Phoebe Harmos, Alfonso J Schmidt, Sally Chappell, Kylie M Price, Ian F Hermans, Franca Ronchese, Graham le Gros, Johannes U Mayer*

Malaghan Institute of Medical Research, Wellington, New Zealand

**Abstract** Single cell isolation from helminth-infected murine intestines has been notoriously difficult, due to the strong anti-parasite type 2 immune responses that drive mucus production, tissue remodeling and immune cell infiltration. Through the systematic optimization of a standard intestinal digestion protocol, we were able to successfully isolate millions of immune cells from the heavily infected duodenum. To validate that these cells gave an accurate representation of intestinal immune responses, we analyzed them using a high-dimensional spectral flow cytometry panel and confirmed our findings by confocal microscopy. Our cell isolation protocol and high-dimensional analysis allowed us to identify many known hallmarks of anti-parasite immune responses throughout the entire course of helminth infection and has the potential to accelerate single-cell discoveries of local helminth immune responses that have previously been unfeasible.

## Introduction

Recent advances in single cell analysis have significantly increased our understanding of multiple diseases and cell types in different tissues (*Svensson et al., 2018*; *Hwang et al., 2018*). However, many of these technologies require single cell suspensions as an input, which limits our assessment of difficult-to-process tissues (*Hwang et al., 2018*; *Nguyen et al., 2018*; *Chen et al., 2018*). One prominent example is the intestine, which is at the center of many research questions that focus on nutrient uptake (*Kiela and Ghishan, 2016*), host-microbiome interactions (*Belkaid and Hand, 2014*; *Tilg et al., 2019*; *Sekirov et al., 2010*), local and systemic immune tolerance (*Worbs et al., 2006*; *Harrison and Powrie, 2013*; *Whibley et al., 2019*) and gastrointestinal diseases and infections (*Mowat and Agace, 2014*; *Sell and Dolan, 2018*; *Fletcher et al., 2013*; *Hendrickson et al., 2002*; *Saleh and Elson, 2011*), but represents a challenging tissue to digest (*Weigmann et al., 2007*; *Reißig et al., 2014*).

The standard digestion procedure to isolate intestinal immune cells located in the small intestinal lamina propria consists of three steps (*Weigmann et al., 2007*; *Reißig et al., 2014*; *Scott et al., 2016*; *Esterházy et al., 2019*) (*Figure 1a*). First the intestinal segment of interest is collected, opened longitudinally to remove its luminal content, washed and cut into small pieces. These pieces then undergo several wash steps with EDTA containing wash buffers at 37°C to remove the epithelial layer and make the lamina propria accessible for enzymatic digestion. Lastly, the tissue is enzymatically digested with collagenases (Collagenase VIII from *Clostridium histolyticum* being among the most popular) and later filtered to obtain a single cell suspension.

While this method results in a high cell yield from steady state intestines, the isolation of cells from severely infected segments remains challenging (*Reißig et al., 2014*; *Scott et al., 2016*; *Webster et al., 2020*) (*Figure 1b*). One of the most prominent examples are infections with intestinal helminths, which represent over 50% of all parasitic infections in human and livestock populations (*McSorley and Maizels, 2012*; *Hotez et al., 2008*; *Jourdan et al., 2018*; *Jackson et al., 2009*). This has limited our analysis of anti-parasite immune responses to imaging approaches, phenotypical

*For correspondence:
jmayer@malaghan.org.nz

Competing interests: The authors declare that no competing interests exist.

**eLife digest** Parasitic worms known as helminths represent an important health problem in large parts of Africa, South America and Asia. Once their larvae enter the body, they head to the gut where they mature into adults and start laying eggs. In areas with poor sanitation, these may then get passed on to other individuals. To defend the body, the immune system sends large numbers of immune cells to the gut, but it usually struggles to eliminate the parasites. Without deworming medication, the infection can last for many years.

Scientists study helminth infections in the laboratory by using worms that naturally infect mice. Understanding exactly how the immune system responds to the infection is essential to grasp why it fails to clear the worms. However, it is difficult to extract immune cells from an infected gut, as the infection creates strong local responses – such as an intense 'slime' production to try to flush out the worms.

The standard procedure to obtain immune cells from the gut consists of three steps: collecting a gut segment and washing it, stripping away the surface layers with chemicals, and finally using enzymes to digest the tissues, which are then filtered to obtain individual cells. However, this protocol is not able to extract cells during infection. Ferrer-Font et al. therefore methodically refined every step of this method, and finally succeeded in obtaining millions of immune cells from infected guts.

For the first time, these cells could then be studied and identified using a new technology called spectral flow cytometry. Over 40 immune cell types were followed throughout the course of infection, revealing that many 'first responders' immune cells were recruited to the gut early on, when the worms were still larvae. However, these cells disappeared once the worms developed into adults. These findings were confirmed by microscopy, which also showed that the first responder cells were found around the developing larvae, likely attacking them. When the adult worms developed, these cells were replaced by other immune cells, which also decreased the longer the worms were present in the gut.

This new extraction process established by Ferrer-Font et al. can also be paired with other technologies that can, for example, reveal which genes are turned on in individual cells. This could help map out exactly how the body fights helminth infections, and how to improve this response. The method could also be useful to extract immune cells from the gut in other challenging scenarios, such food allergies or inflammatory bowel disorders.

observations in transgenic mouse strains or the assessment of secondary locations like the draining lymph nodes, blood or spleen (*McSorley and Maizels, 2012*; *Maizels and McSorley, 2016*; *Mishra et al., 2014*), which might only partially reflect local immunity. As helminth infections are strongly linked to chronic impairments that affect nutrition availability (*Koski and Scott, 2001*; *Crompton and Nesheim, 2002*); memory, cognition and physical development (*Ezeamama et al., 2005*; *Pabalan et al., 2018*; *Nokes et al., 1992*); changes in the microbiota (*Gause and Maizels, 2016*; *Ramanan et al., 2016*) and modulation of local and systemic immunity (*Mishra et al., 2014*; *Maizels et al., 2009*), an optimized digestion protocol is needed to further investigate the infected intestinal tissue.

## Results and discussion

Difficulties with intestinal digests during helminth infection have been associated with a strong anti-parasite type 2 immune response that drives mucus production (*Hashimoto et al., 2009*; *von Moltke et al., 2016*), alters the epithelium (*Gerbe et al., 2016*; *Howitt et al., 2016*), induces immune cell infiltration (*Inclan-Rico and Siracusa, 2018*) and causes tissue remodelling (*Motran et al., 2018*; *Boyett and Hsieh, 2014*) (*Figure 1c*). In order to investigate a model of both acute and chronic helminth infection, we infected C57BL/6 mice with *Heligmosomoides polygyrus bakeri* (also known as *Heligmosomoides bakeri*; *Behnke and Harris, 2010*), a naturally occurring rodent parasite with an exclusive intestinal life cycle (*Monroy and Enriquez, 1992*; *Reynolds et al., 2012*). Infective L3 larvae penetrate the intestinal tissue of the duodenum within 24 hr of ingestion, undergo larval development in the muscularis externa and return to the lumen within 10 days post

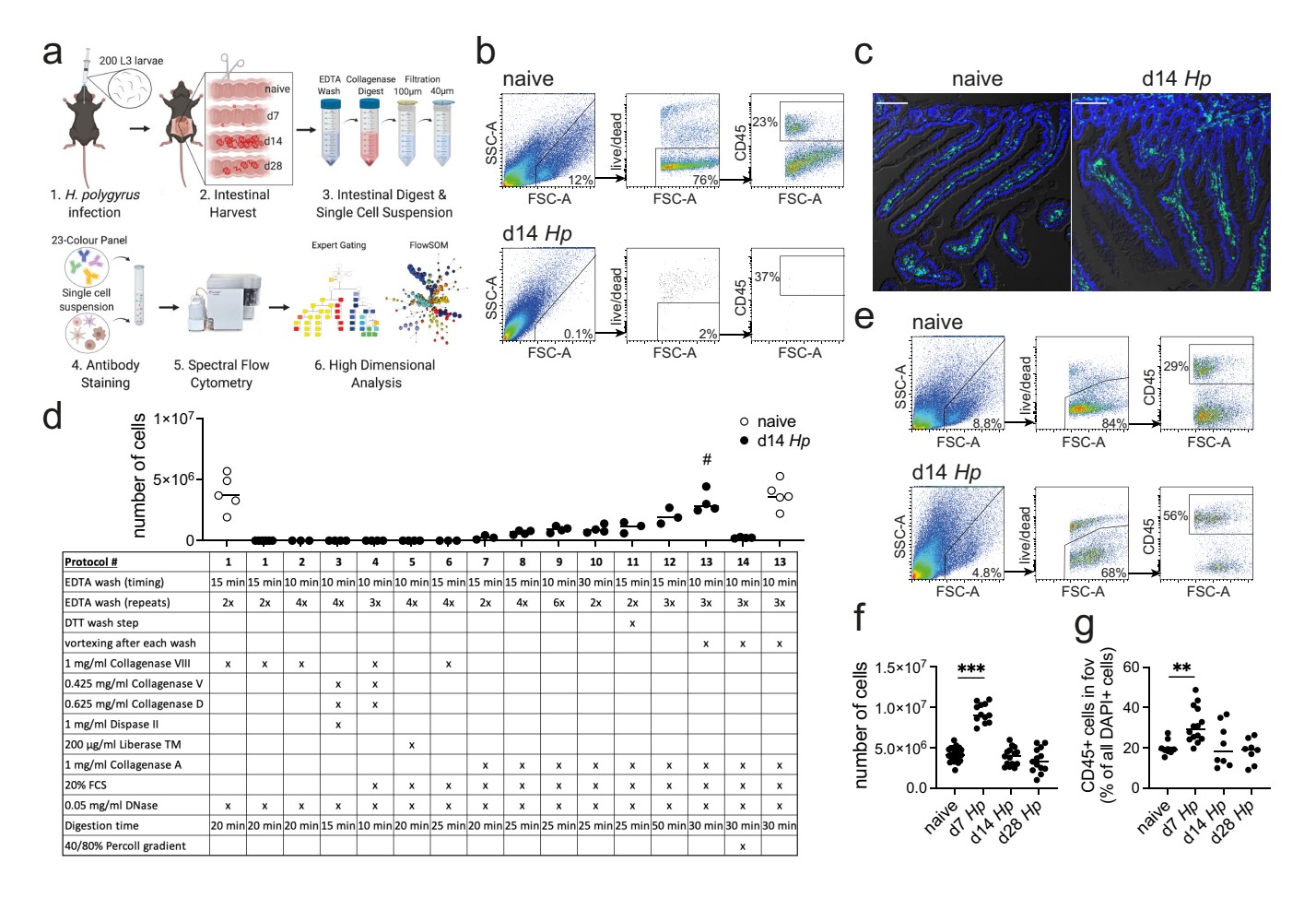

**Figure 1.** Optimization of a standard intestinal digestion protocol for the heavily infected duodenum. (**a**) Schematic of a general intestinal digestion protocol (created with biorender.com). (**b**) Digest of naïve and day 14 hr. *polygyrus (Hp)*-infected duodenal segments using the standard digestion protocol. (**c**) Intestinal cryosections stained with CD45-FITC (green) and DAPI (blue) from naïve and day 14 infected intestines. Scale bar = 100 µm (representative of >10 sections from 3 to 5 mice per group and two independent experiments). (**d**) Number of live cells isolated from naïve or day 14 infected duodenal segments during the systematic optimization of the standard digestion protocol. Further details can be found in *Figure 1—figure supplement 1* and *Figure 1—figure supplement 2* (n = 3–5 samples per group, combined data from at least two independent experiments; # depicts the digestion protocol that yielded comparable cell numbers between naïve and infected samples, all other protocols showed a significant difference to the naïve group when compared by ordinary one-way ANOVA followed by Holm-Sidak's multiple comparisons test). (**e**) Digest of naïve and day 14 infected duodenal segments using the optimised *Hp* digestion protocol (#13). (**f**) Number of live cells isolated from naïve, day 7, day 14 and day 28 infected duodenal segments using the optimized *Hp* digestion protocol (n > 12 samples per group, combined data from at least three independent experiments; Kruskal-Wallis followed by Dunn's multiple comparisons test compared to the naïve group; ***p≤0.001). (**g**) Quantification of CD45+ cells present in the field of view (fov, 635.90µm x 635.90µm) in cryosections from the same timepoints (representative of >10 sections from 3 to 5 mice per group from two independent experiments; Kruskal-Wallis followed by Dunn's multiple comparisons test compared to the naïve group; **p≤0.01). The online version of this article includes the following figure supplement(s) for figure 1:

**Figure supplement 1.** Modification of a standard intestinal digestion protocol to isolate single cells from heavily infected duodenal segments.
**Figure supplement 2.** Further optimization of a single cell isolation protocol from heavily infected duodenal segments based on Collagenase A digestion.
**Figure supplement 3.** Intestines from *H. polygyrus*-infected Stat6ko mice can be digested with the standard cell isolation protocol.
**Figure supplement 4.** Assessment of epitope integrity of digested and non-digested splenocytes.
**Figure supplement 5.** Comparison of different commercial intracellular staining kits on digested lamina propria cells.

infection, where the adult worms mate and develop a chronic infection in C57BL/6 mice (*Reynolds et al., 2012*; *Smith et al., 2016*). The peak of acute immunity is usually studied around day 14 post infection and we focused on this time point and the heavily infected duodenum (*Filbey et al., 2014*; *Elliott et al., 2008*), to optimize our digestion protocol.

In order to develop a digestion protocol for heavily infected intestines, we followed a systematic approach and optimized each step of the standard intestinal digestion protocol. First, we modified the EDTA wash steps to remove the increased amount of mucus but did not observe an improvement in cell yield (*Figure 1d* and *Figure 1—figure supplement 1*; digestion protocols #1, 2, 6). This was followed by testing a variety of collagenases that have been reported for intestinal digests, as we hypothesized that the intestinal remodeling that occurred during helminth infection could negatively impact the digestion procedure. Indeed, we found that Collagenase A from *Clostridium histolyticum* (*Figure 1d* and *Figure 1—figure supplement 1*; digestion protocols #7, 8), but not Collagenase VIII, Collagenase D, Dispase or Liberase TM (*Figure 1d* and *Figure 1—figure supplement 1*; digestion protocols #3–5), showed an increase in cell yield when used in conjunction with the standard digestion protocol.

To further optimize the protocol, we focused on Collagenase A-based digestion and increased and modified the wash steps and observed a further increase in cell yield (*Figure 1d* and *Figure 1—figure supplement 2*; digestion protocols #9–12). Importantly, strong vortexing after each wash step significantly improved the outcome of digestion (*Figure 1d* and *Figure 1—figure supplement 2*; digestion protocol #13), suggesting that the epithelium is harder to remove in helminth-infected tissues. Indeed, observations from Stat6ko mice confirmed that the physiological changes that impair the intestinal digest using the standard protocol, were all linked to type two immune responses, as intestines from infected Stat6ko mice could readily be digested (*Figure 1—figure supplement 3*). We also assessed intra-epithelial cells in the EDTA wash, but could not detect any CD45 + cells in preparations from infected animals, emphasizing that our protocol should only be used to isolate lamina propria cells. Several intestinal cell isolation protocols also utilize a final gradient centrifugation step to further isolate immune cells (*Weigmann et al., 2007*; *Esterházy et al., 2019*). However, in our hands this resulted in a dramatic drop in cell yield and was therefore omitted (*Figure 1d* and *Figure 1—figure supplement 2*; digestion protocol #14). Our optimized lamina propria cell isolation protocol for *H. polygyrus*-infected intestines thus included three 10 min 2 mM EDTA wash steps (each followed by vigorous vortexing) and a 30 min digest with 1 mg/ml Collagenase A, 20% FCS and 0.05 mg/ml DNase (see Appendix 1 for step-by-step instructions).

When we compared intestinal digests from naïve animals using the standard or optimized cell isolation protocol, we observed highly comparable outcomes (*Figure 1d*; digestion protocols #1 and 13). Both digestion protocols resulted in a cell yield of 3–6 million live cells per duodenum with 70–80% viability and 20–30% frequency of CD45+ cells. To assess the effectiveness of our digestion protocol during the different stages of *H. polygyrus* infection, we harvested the duodenum from naïve C57BL/6 mice and at day 7, day 14 and day 28 of *H. polygyrus* infection, which represented time points of larval development in the muscularis externa, as well as acute and chronic adult worm infection, respectively. We observed that samples from all time points could be successfully digested using our optimized digestion protocol and that duodenal digests from 14- and 28 days post infection yielded 3–6 million live cells per sample (*Figure 1e,f*). We furthermore observed a consistent doubling of the cell count to 8–11 million live cells per duodenum at day seven post infection and observed a similar trend when we quantified CD45+ cells in cryosections from these time points (*Figure 1f,g*).

To understand these differences and validate whether our protocol was suitable for subsequent single cell analysis and immunophenotyping, we characterized the isolated cells further using a 23-color spectral flow cytometry panel that incorporated many of the hallmark surface and intracellular markers for type two immune responses that have been associated with helminth infections (*Maizels and McSorley, 2016*; *Reynolds et al., 2012*) (see *Supplementary file 1* for details regarding markers, fluorophores, clones and staining concentrations used). Our staining panel was designed to identify both innate and adaptive immune cell populations and allowed us to assess eosinophils, neutrophils, different subsets of monocytes, macrophages and dendritic cells, as well as the three main populations of innate lymphoid cells (ILC1, ILC2, and ILC3) and effector T cells populations (Th1, Th2, Th17), T regulatory cells and B cells as well as their proliferation through ki67 expression within the same panel.

To guarantee optimal staining conditions, we tested our optimized digestion protocol on splenocytes and compared digested to non-digested cells, as collagenase digests can negatively affect surface epitope integrity. While we observed a reduction in the MFIs of several markers (namely Ly6G, MHCII, CD45 and CD127), all positive stained cell populations could be clearly identified (*Figure 1—figure supplement 4*). Isolated intestinal lamina propria cells also proved a challenge for intracellular staining, as different commercial intracellular staining kits significantly affected the cellular, but not debris, scatter profiles and varied in the resolution of intracellularly antibody staining (*Figure 1—figure supplement 5*). In our hands, the eBioscience FoxP3/Transcription Factor Staining Buffer Set yielded the best results and was used henceforth.

We isolated immune cells from the three main stages of *H. polygyrus* infection (day 7, day 14 and day 28), representing larval development, as well as acute and chronic worm infection and used a combination of high-dimensional analysis tools and manual gating strategies to assess changes within each immune cell population (*Figure 2a* and *Figure 2—figure supplement 1* and *Figure 2—figure supplement 2*). In line with previous findings (*Inclan-Rico and Siracusa, 2018*), we observed a strong infiltration of immune cells such as neutrophils and monocytes at day 7 post infection, which we verified by confocal microscopy and were primarily localized around the developing larvae explaining the increase in total cell number at this timepoint (*Figure 2a–c* and *Figure 2—figure supplement 3*). At later time points this inflammatory response receded, which was likely linked to the worms exiting the intestinal tissue and inhabiting the lumen. Peak expression of RELMα in resident macrophages, which is a hallmark for their alternative activation and wound repair responses (*Esser-von Bieren et al., 2013*; *Krljanac et al., 2019*), was observed at day 14 and was again localized within the granulomas (*Figure 2b,c* and *Figure 2—figure supplement 3*). While type two innate lymphoid cells did not increase in frequency over time, ki67 expression increased, suggesting cell proliferation and activation (*Figure 2d*), as previously described (*von Moltke et al., 2016*; *Schneider et al., 2018*). GATA3+ Th2 cells, important drivers of type two immunity (*Reynolds et al., 2012*; *Mohrs et al., 2005*), were detected throughout all stages of infection, increased in frequency over time and showed high ki67 expression (*Figure 2b*). Interestingly, ki67 expression strongly decreased at day 28 post infection for all cell types analyzed (*Figure 2d* and *Figure 2—figure supplement 4*), which could be linked to the strong immunomodulatory properties reported during chronic worm infection (*Maizels and McSorley, 2016*; *Grainger et al., 2010*) (*Figure 2e*).

To validate that our cell isolation protocol resulted in an accurate representation of intestinal immune responses, we quantified B220+, Siglec F+, CD3+ CD4+, CD3+ CD4- and CD64+ cells using confocal microscopy (*Figure 2—figure supplement 5a–c*) and compared their frequency to our spectral flow cytometry data. We observed that the frequencies of B220+, CD3+ CD4+, and CD64+ cells were highly comparable between confocal microscopy and spectral flow cytometry, while Siglec F+ cells were overrepresented in our spectral flow cytometry data and CD3+ CD4- cells were underrepresented. However, changes within immune cell populations at the peak of *H. polygyrus* infection were faithfully reported by both confocal microscopy and spectral flow cytometry (*Figure 2—figure supplement 5d*), emphasizing that cell ratios defined by single cell analysis need to be carefully validated within the tissue before conclusions are drawn.

Another important conclusion from our analysis was that the strong inflammatory immune responses that we had observed during *H. polygyrus* development in the muscularis externa, were specific to the infected tissue and were not observed to the same extent in the draining lymph nodes. Furthermore, ratios of ILC populations and T helper subsets were also strikingly different between the lamina propria and the draining lymph nodes at steady state, as were their proliferation kinetics and changes in proportion throughout the course of *H. polygyrus* infection (*Figure 2—figure supplement 6* and *Figure 2—figure supplement 7*).

To highlight the potential of our protocol for future studies that utilize current single-cell analysis tools, such as single cell RNA sequencing, we assessed the RNA quality of purified B cells, CD4 T cells and macrophages isolated from naïve or day 14 *hr. polygyrus*-infected mice (*Figure 2—figure supplement 8a,b*). Our RNA quality analysis using the Agilent TapeStation resulted in high RIN numbers (range 6.8–10 for naïve and 7.0–10 for day 14 *hr. polygyrus* samples) (*Figure 2—figure supplement 8e*). However, no clear separation of the 18S and 28S peaks could be observed on the gel image or the electropherograms (*Figure 2—figure supplement 8c,d*). While the RIN numbers might not have been correctly calculated, and a technical optimization of the TapeStation protocol might be necessary, no RNA degradation was observed in naïve and day 14 *hr. polygyrus* samples,

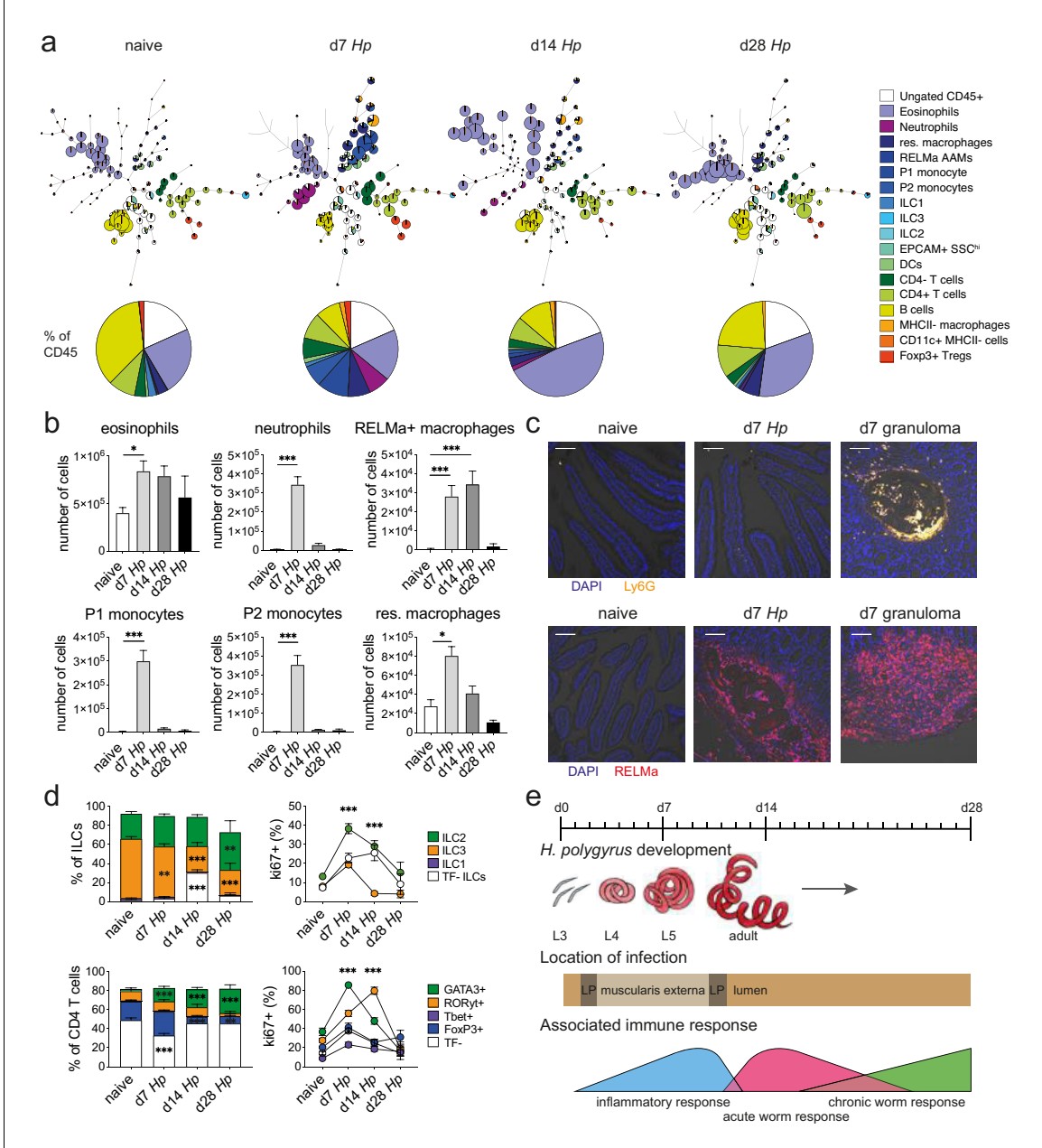

**Figure 2.** Spectral flow cytometric analysis of isolated intestinal immune cells during the course of *H. polygyrus* infection. (**a**) FlowSOM (top) and manual (bottom) analysis of live CD45+ cells isolated from naïve, day 7, day 14 and day 28 infected duodenal segments stained with 23 surface and intracellular antibodies and gated as described in *Figure 2—figure supplement 1* (n = 3–8 samples per group, combined data from two independent experiments). (**b**) Quantification of different innate immune cell populations during the course of *H. polygyrus* infection (mean ± s.e.m.; Kruskal-Wallis followed by Dunn's multiple comparisons test compared to the naïve group; *p≤0.05, ***p≤0.001). (**c**) Representative images from intestinal cryosections stained with Ly6G-PECF594 (orange) and DAPI (blue) from naïve and day 7 infected duodenal segments (top) or stained with RELMα-APC (red) and DAPI (blue) from naïve, day 7 and day 14 infected duodenal segments (bottom). Scale bar = 100 μm (representative of >10 sections from 3 to 5 mice per group and two independent experiments). (**d**) Proportions of ILC and CD4 T cell populations and their expression of the proliferation marker ki67 during the course of infection (mean ± s.e.m.; 2-way ANOVA followed by Dunnett's multiple comparisons test compared to each of the naïve groups (stacked bar graphs) or compared to the combined naïve group (line graphs); **p≤0.01, ***p≤0.001). (**e**) Schematic of *H. polygyrus* development, location and associated immune responses during the course of infection.

The online version of this article includes the following figure supplement(s) for figure 2:

**Figure supplement 1.** Gating strategy for duodenal lamina propria cells.

**Figure supplement 2.** FlowSOM analysis of duodenal lamina propria cells.

**Figure supplement 3.** Intestinal cryosections highlight infiltration of innate immune cells within larval granulomas.

*Figure 2 continued on next page*

*Figure 2 continued*

**Figure supplement 4.** Analysis of isolated intestinal immune cells during the course of *H. polygyrus* infection.
**Figure supplement 5.** Quantification of intestinal immune cells detected by confocal microscopy or spectral flow cytometry.
**Figure supplement 6.** Gating strategy for mesenteric lymph node cells.
**Figure supplement 7.** Spectral flow cytometric analysis of mesenteric lymph node cells during the course of *H. polygyrus* infection.
**Figure supplement 8.** Assessment of RNA quality of sorted intestinal immune cells from naïve and *H. polygyrus*-infected mice.

suggesting that the extraction of high-quality RNA is feasible from both naïve and day 14 hr. *polygyrus*-infected mice using our cell isolation protocol.

Thus, our cell isolation protocol and high-dimensional analysis allowed us to characterize many known hallmarks of innate and adaptive anti-parasite immune responses throughout the entire course of helminth infection. We were able to validate these changes using confocal microscopy and while we could observe differences in the reported cell ratios, changes between naïve and infected samples were faithfully reported by both approaches.

Importantly, many of these changes were only observed locally, highlighting the requirement for good cell isolation techniques to investigate intestinal responses against helminths directly.

In addition to flow cytometric immunophenotyping, we were also able to extract high-quality RNA from cells isolated with our protocol, which could accelerate single-cell discoveries of local helminth immune responses through current single-cell analysis tools, such as single cell RNA sequencing, which has previously been unfeasible.

## Materials and methods

### Ethics statement

All animal experiments were carried out at the Malaghan Institute of Medical Research, were approved by the Victoria University of Wellington Animal Ethics Committee (permit 24432) and carried out according to institutional guidelines.

### Mice

C57BL/6 and Stat6ko (*B6.129S2(C)-Stat6tm1Gru/J*) mice were imported from The Jackson Laboratory and bred at the Malaghan Institute of Medical Research, Wellington, New Zealand. Mice were housed under specific pathogen free conditions and age-matched female adult animals were used in each experiment.

### *Heligmosomoides polygyrus* infection

The *H. polygyrus* life cycle was maintained as previously described (*Johnston et al., 2015*). For experimental infections, mice were infected with 200 L3 larvae by oral gavage at 6–8 weeks of age and intestines and draining lymph nodes were harvested at the indicated time points. Adult worm burden was quantified by mounting opened intestines inside a 50 ml falcon filled with PBS. After 3 hr at 37°C, worms were collected from the bottom of the tube and counted under a microscope.

### Cell isolation

Lamina propria cells were isolated from the first 8 cm of intestine according to isolation protocols described in this manuscript. Optimal digestion was achieved when intestinal segments were excised, cleaned and cut into small pieces. Samples were then washed with 2 mM EDTA/HBSS (Gibco) three times for 10 min at 37° C and 200 rpm in a shaking incubator, followed by three pulse vortexing steps at 2500 rpm (maximum speed) for 3 s after each incubation. After the final EDTA wash step, samples were digested in 10 ml RPMI (Gibco) containing 20% FBS (Gibco), 1 mg/ml Collagenase A (Roche #10103578001, 0.223 U/mg solid) and 0.05 mg/ml DNAse (Roche #10104159001, 2916 Kunitz units/mgL) for 30 min at 37°C and 200 rpm in a shaking incubator, with vigorous manual shaking every 5 min. Digestion was quenched with FACS buffer and samples were passed through a 100 µm and 40 µm cell strainer to obtain a single cell suspension. An illustrated step-by-step protocol describing the procedure can be found in Appendix 1.

Individual duodenum draining mesenteric lymph nodes were identified as the most proximal lymph nodes of the mesenteric lymph node chain (*Esterházy et al., 2019*; *Mayer et al., 2017*), and were digested with 100 µg/mL Liberase TL and 100 µg/mL DNase I (both from Roche, Germany) for 30 min at 37°C and passed through a 70 µm cell strainer.

## Conventional and spectral flow cytometry

For conventional flow cytometry cells were resuspended in 0.5 ml of 20 µg/ml DNase containing FACS buffer, stained with DAPI to identify dead cells, filtered and analyzed using a BD LSRFortessa SORP flow cytometer. For spectral flow cytometry, intestinal and lymph node samples were washed in 200 µL FACS buffer and incubated with Zombie NIR Fixable Viability dye (Biolegend) for 15 min at room temperature. After washing, cells were incubated with Fc block (clone 2.4G2, affinity purified from hybridoma culture supernatant) for 10 min followed by the incubation of surface antibodies (see *Supplementary file 1*) for 25 min at 4°C in the presence of 20 µg/ml DNase and Brilliant Buffer Plus (BD Biosciences). Cells were fixed and permeabilized with the FoxP3/Transcription Factor Staining Buffer Set (eBioscience) according to manufacturer's instructions and incubated with intracellular antibodies (see *Supplementary file 1*) for 45 min at 4°C. Cells were then resuspended in FACS buffer, filtered, and analyzed on a 3-laser Aurora spectral flow cytometer (Cytek Biosciences).

## Data analysis

FCS files were manually analyzed using FlowJo (v10.6, Tree Star) or evaluated with high-dimensional data analysis tools using Cytobank (v7.2, Cytobank Inc). After compensation correction in FlowJo, single, live, CD45+ events were imported into Cytobank and transformed to arcsinh scales. FlowSOM analysis was performed on 1,200,000 concatenated lamina propria and 1,000,000 concatenated lymph node cells, with an equal distribution of samples. Different cluster analyses were performed and 121 clusters were identified as the most representative for both data sets.

## Imaging

For histological sections, 5 µm FFPE (Formalin fixed paraffin embedded) sections were stained using a standard H and E protocol (*Jacobson et al., 2008*) and visualized using a BX51 microscope (Olympus) equipped with a 10X NA 0.3 objective. For confocal microscopy, samples were processed and stained using a standard immunofluorescence protocol (*Schmidt et al., 2019*). Briefly, 1 cm long pieces of intestine were fixed in 4% PFA for 1 hr, incubated in 20% sucrose overnight and rinsed in PBS. Samples were then snap-frozen in OCT compound (Tissue-Tek) using a Stand-Alone Gentle Jane Snap-freezing system (Leica Biosystems). Cryosections of 7 µm were blocked with Fc Block (clone 2.4G2, affinity purified from hybridoma culture supernatant) for 1 hr and stained with CD45-FITC (clone 30-F11, Biolegend), CD64-PE (clone X54-5/7.1, Biolegend), Ly6G-PECF594 (clone 1A8, Biolegend), RELMα-APC (clone DSBRELM, eBioscience), B220-PECF594 (clone RA3-6B2, BD Biosciences), CD3-PE (clone 145–2 C11, eBioscience) or CD4-APC (clone RM4-5, BD Biosciences) for 1 hr. For nuclear staining, sections were incubated with DAPI (2 µg/ml) for 10 min. Images were taken with an inverted IX 83 microscope equipped with a FV1200 confocal head (Olympus) using a 20X, N. A 0.75 objective. Images were acquired using the FV10-ASW software (v4.2b, Olympus) and analyzed with ImageJ (*Schindelin et al., 2012*) (v1.52n). Image quantification analysis per field of view was performed using CellProfiler (*Lamprecht et al., 2007*) (v3.1.8) and based on the spatial co-expression of immune cell markers and DAPI-positive nuclei. Cell quantification per villi section per mm (*Hwang et al., 2018*) was based on manual selection of the villi and electronic quantification of the area and positively stained cells.

## Cell sorting, RNA extraction and RNA quality assessment

Single cell suspensions from naïve or day 14 hr. *polygyrus*-infected C57BL/6 mice were stained with CD45-BUV395 (clone 30-F11, BD Biosciences), CD64-Al647 (clone X54-5/7.1, Biolegend), MHCII-PE (clone M5/114.15.2, BD Biosciences), CD19-BB515 (clone 1D3, BD Biosciences), CD3-BV605 (clone 17A2, Biolegend), CD4-Pac Blue (clone RM4-5, BD Biosciences) and DAPI. 700,000 CD45+ cells were sorted into FACS buffer using a BD Influx cell sorter (BD Biosciences) followed by purification of B cells, CD4 T cells and macrophages. 5,000 cells of each population were sorted into 100 µl RNA lysis buffer (Zymo Research) and stored at −80C. RNA was extracted using the Quick-RNA

MicroPrep Kit (Zymo Research) and its quality assessed using the High Sensitivity RNA Screen Tape (Agilent) and a 4150 TapeStation System (Agilent) according to the manufacturer's instructions.

## Statistical analysis

Experimental group sizes ranging from 3 to 5 animals were chosen to ensure that a two-fold difference between means could be detected with a power of at least 80%. Prism 6 Software (GraphPad) was used to calculate the s.e.m. and the statistical differences between groups and samples for each data set as detailed in the corresponding figure legends, with $p \leq 0.05$ being considered as significant.

## Source data

All spectral flow cytometry data sets presented in this study can be downloaded from flowrepository (http://flowrepository.org/id/FR-FCM-Z28B).

## Acknowledgements

This work was supported by an IRO grant from the Health Research Council of New Zealand to the Malaghan Institute and an 'In Aid of Research' grant from the Research For Life foundation to JUM. PH was supported by the Maurice Wilkins Centre for Molecular Biodiscovery. LFF, AJS, SCC and KMP were supported by the Hugh Green Foundation. We would also like to acknowledge the staff of the Biomedical Research Unit for their assistance.

## Additional information

### Author ORCIDs

Johannes U Mayer (iD) https://orcid.org/0000-0001-6225-7803

### Ethics

Animal experimentation: All animal experiments were carried out at the Malaghan Institute of Medical Research, were approved by the Victoria University of Wellington Animal Ethics Committee (permit 24432) and carried out according to institutional guidelines.

### Funding

| Funder | Grant reference number | Author |
| --- | --- | --- |
| Research for Life foundation | In Aid of Research grant | Johannes U Mayer |
| Hugh Green Foundation | | Laura Ferrer-Font |
| Hugh Green Foundation | | Alfonso J Schmidt |
| Hugh Green Foundation | | Sally Chappell |
| Hugh Green Foundation | | Kylie M Price |
| Maurice Wilkins Centre | | Phoebe Harmos |
| Maurice Wilkins Centre | | Ian F Hermans |
| Health Research Council of New Zealand | IRO grant | Laura Ferrer-Font |
| Health Research Council of New Zealand | IRO grant | Palak Mehta |
| Health Research Council of New Zealand | IRO grant | Phoebe Harmos |
| Health Research Council of New Zealand | IRO grant | Alfonso J Schmidt |
| Health Research Council of New Zealand | IRO grant | Sally Chappell |

| | | |
|---|---|---|
| Health Research Council of New Zealand | IRO grant | Kylie M Price |
| Health Research Council of New Zealand | IRO grant | Ian F Hermans |
| Health Research Council of New Zealand | IRO grant | Franca Ronchese |
| Health Research Council of New Zealand | IRO grant | Graham le Gros |
| Health Research Council of New Zealand | IRO grant | Johannes U Mayer |

The funders had no role in study design, data collection and interpretation, or the decision to submit the work for publication.

### Author contributions
Laura Ferrer-Font, Conceptualization; Formal analysis; Investigation; Visualization; Methodology; Writing - review and editing; Palak Mehta, Phoebe Harmos, Formal analysis, Investigation, Methodology, Writing - review and editing; Alfonso J Schmidt, Investigation, Visualization, Methodology, Writing - review and editing; Sally Chappell, Investigation, Methodology, Writing - review and editing; Kylie M Price, Ian F Hermans, Franca Ronchese, Graham le Gros, Supervision, Funding acquisition, Writing - review and editing; Johannes U Mayer, Conceptualization, Formal analysis, Supervision, Funding acquisition, Investigation, Visualization, Methodology, Writing - original draft, Writing - review and editing

### Decision letter and Author response
Decision letter https://doi.org/10.7554/eLife.51678.sa1
Author response https://doi.org/10.7554/eLife.51678.sa2

## Additional files

### Supplementary files
• Supplementary file 1. 23-color spectral flow cytometry panel for the analysis of intestinal immune cells during helminth infection. Antibodies used for the high-dimensional analysis of intestinal immune cells are listed by channel, emission, marker, fluorochrome, clone, company, catalog ID, optimal staining dilution and working concentration.

• Transparent reporting form

### Data availability
Lamina propria and lymph node data sets can be downloaded from flowrepository (http://flowrepository.org/id/FR-FCM-Z28B).

The following dataset was generated:

| Author(s) | Year | Dataset title | Dataset URL | Database and Identifier |
|---|---|---|---|---|
| Ferrer-Font L, Mayer JU | 2019 | Lamina propia and MLN cells during *H. polygyrus* infection | http://flowrepository.org/id/FR-FCM-Z28B | FlowRepository, FR-FCM-Z28B |

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

**Appendix 1**

## Step-by-step lamina propria cell isolation protocol for *H. polygyrus*- infected intestines
according to Ferrer-Font et. al. 2020

### Reagents
- RPMI (Gibco RPMI Medium 1640 #11875–093)
- HBSS (Gibco Hank's Balanced Salt Solution #14175–095)
- DPBS (Gibco Dulbecco's Phosphate Buffered Saline #14190–144)
- UltraPure 0.5M EDTA (Thermofisher #15575020)
- DNAse I (Roche #10104159001, 2916 Kunitz units/mgL)
- Collagenase A from *Clostridium histolyticum* (Roche #10103578001, 0.223 U/mg solid)
- Fetal Bovine Serum (Gibco)
- FACS buffer (PBS, 2% FBS, 2 mM EDTA)
- EDTA wash buffer (HBSS, 2 mM EDTA)
- Wash buffer (HBSS)
- Collection buffer (HBSS, 2% FBS)
- Digestion mix (RPMI, 20% FBS, 1 mg/ml Collagenase A, 0.05 mg/ml DNAse)
- Trypan Blue Solution (Gibco)

### Materials
- Funnels
- 10 × 10 cm Gauze (140 µm mesh size)
- Tweezers to hold gauze in place
- 50 ml falcon tubes
- 40 µm cell strainers
- 100 µm cell strainer
- 25 ml serological pipette
- Scissor/tweezers for collection
- Petri dish
- Haemocytometer

### Equipment
- Shaking incubator
- Centrifuge

### Step-by-step protocol
1. Warm bottles of RPMI, HBSS and HBSS/2 mM EDTA to 37°C.
2. Prepare 50 ml falcon tubes with 10 ml HBSS/2% FBS for each sample (keep on ice). Take petri dish, PBS and scissors/tweezers for collection of samples.
3. Sacrifice mouse, spray with EtOH and perform midline incision of the skin and muscle layer (*Appendix 1—figure 1A*).
4. Expose the intestines and excise the entire small intestine or a segment of interest and separate it from mesentery/fat tissue (*Appendix 1—figure 1B-D*).
5. Place intestinal segment on moist paper and remove the Peyer's patches.
6. Cut the segment longitudinally (*Appendix 1—figure 1 E*).
7. Remove worms and intestinal content (*Appendix 1—figure 1F and G*).
8. Wash intestinal segment in PBS and cut into small pieces (~5 mm long) (*Appendix 1—figure 1H and I*).
9. Collect pieces in 50 ml falcon tube containing 10 ml HBSS/FBS, shake well and keep on ice (*Appendix 1—figure 1J*).
10. Repeat steps 3–8 for remaining samples.
11. Prepare 10 ml fresh digestion mix for each sample containing 20% FBS, 1 mg/ml Collagenase A and 0.05 mg/ml DNAse in RPMI and warm up to 37° C.
12. Filter each sample through a 10 × 10 cm gauze (140 µm mesh size) placed on top of a funnel. Discard the flow-through (*Appendix 1—figure 1K*).
    The same gauze can be reused in each wash step.

13. Wash sample with 10 ml of warm HBSS twice. Discard the flow-through (*Appendix 1—figure 1I*).
14. Remove the gauze from the funnel and collect sample in a 50 ml falcon tube containing 10 ml HBSS/EDTA (*Appendix 1—figure 1M-O*).
    It is advisable to process no more than 4–5 samples in parallel.
15. Incubate samples for 10 min at 37° C and 200 rpm in a shaking incubator (*Appendix 1—figure 1P*).
16. Pulse vortex samples three times at 2500 rpm (maximum speed) for 3 s after each incubation.
    A cloudy suspension should be observed.
17. Repeat wash steps 12–16 two more times, for a total of three 10 min wash steps. The HBSS/EDTA suspension should become less cloudy with each wash step. If significant amounts of debris are still observed after the last wash add a forth wash step.
18. After the final EDTA wash step, filter each sample through a 10 × 10 cm gauze and wash with 10 ml of warm HBSS twice.
19. Remove the gauze from the funnel and collect sample in a 50 ml falcon tube containing 10 ml digestion mix.
20. Incubate samples for 30 min at 37°C and 200 rpm in a shaking incubator. Shake samples vigorously every 5 min.
    The pieces will not be fully digested at this time point but have been optimised to yield the highest cell number with the least effect on viability and epitope integrity.
21. Add 10 ml of FACS buffer to each sample and keep on ice to stop the digestion.
22. Filter each sample through a 100 µm and 40 µm cell strainer into a new 50 ml falcon tube using a 25 ml serological pipette and place on ice (*Appendix 1—figure 1Q and R*).
    Do not simply pipette the sample on top of the strainer but force it through the mesh with pressure from the pipette controller.
23. Centrifuge the single cell suspension at 600 x *g* for 6 min at 4°C.
24. Discard the supernatant and resuspend in 1 ml of 20 µg/ml DNAse containing FACS buffer.
    Use DNAse containing FACS buffer in all downstream procedures to reduce clumping.
25. Count live cells in a 1:1 suspension of Trypan Blue using a haemocytometer.
26. Process desired number of cells for flow cytometry or other downstream applications.
    Lamina propria digests contain a high percentage of cell debris, which will stain with viability dyes. Optimise the concentration of viability dye and record samples using a high Scatter threshold to facilitate the identification of cells. If cells are to be sorted, pre-enrich the samples by pre-sorting CD45+ cells or using positive selection beads. Avoid enrichment systems that are easily clogged.

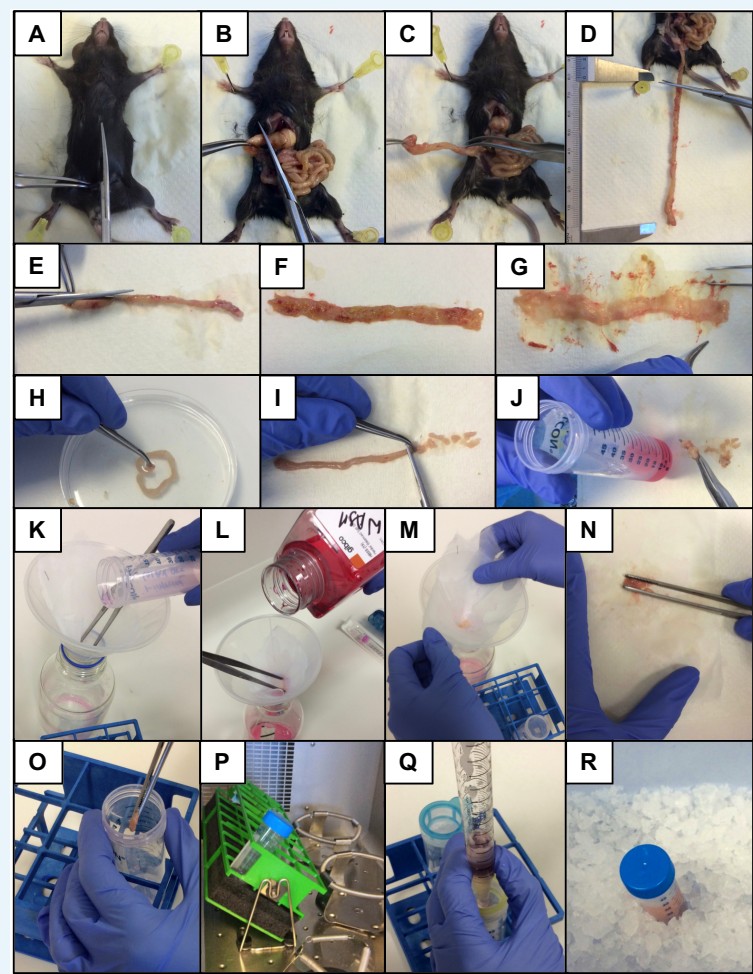

**Appendix 1—figure 1.** Illustrations of the step-by-step lamina propria cell isolation protocol for *H. polygyrus*-infected intestines. Pictures illustrating the different steps required for the lamina propria single-cell isolation from *H. polygyrus*-infected intestines are shown.

