## [Decision Letter]

Thank you for submitting your article "Single-cell analysis of intestinal immune cells during helminth infection" for consideration by *eLife*. Your article has been reviewed by three peer reviewers, including Nicola L Harris as the Reviewing Editor and Reviewer #1, and the evaluation has been overseen by Satyajit Rath as the Senior Editor. The following individuals involved in review of your submission have agreed to reveal their identity: John Grainger (Reviewer #2); Lisa Reynolds (Reviewer #3).

The reviewers have discussed the reviews with one another and the Reviewing Editor has drafted this decision to help you prepare a revised submission.

Summary:

All reviewers agree that the manuscript by Ferrer-Font et al. has the potential to accelerate discovery in the area of local helminth immune responses by reporting what is a significant methodological advance for the field. This advance, which the authors have enabled through careful comparison of a large variety of digestion techniques, will enable others in the field to investigate the intestinal responses against helminths directly, enabling important experimental tools, including single cell sequencing analysis, to be enacted where such approaches were previously unfeasible.

However, it does not offer, in its present form, any major new biological insight. Thus all three reviewers strongly recommended it is re-submitted as a tools and resources manuscript.

They also agreed the manuscripts main point – to provide a tool to allow single cell analysis – needs to be substantiated further by addressing the following points in a revision:

Essential revisions:

Experiment 1) The reviewers agree that a more quantitative comparison of their flow results to histology would be very valuable to confirm how representative their flow results are in reflecting cell types present in the tissue. It would be sufficient to provide this for day 14 (using specific markers for eosinophils, CD4^+^ or CD4- lymphoid cells, B cells and macrophages). The reviewers acknowledge that no digestion protocol will be able to perfectly preserve cell ratios present in the tissue, but it would be valuable for the reader to have this information and to clearly state in the manuscript the caveats of using single cell analysis to describe cell ratios. Please note that quantitative graphs (cell counts or pixel area if positive staining per mm^2^ of tissue, using a set number of villi and tissue sections) should be provided.

Experiment 2) To substantiate the claim that this techniques will be useful for single cell analysis it would be useful for the reader to know whether it can be employed for single cell RNA seq experiments. In the regard the authors should provide an indication of RNA quality able to be obtained from distinct cell populations harvested from the day 14 post-infection digest. Note that the reviewers acknowledge high quality RNA is difficult to obtain from neutrophils and eosinophils, thus it would be sufficient if the authors provided data for the other major cell subsets namely macrophages (CD64+MHCII+), T cells and B cells.

Experiment 3) I would be very useful if the authors collected the cells released from the epithelial layer (intra-epithelial cells) during the EDTA and vortexing stages and subjected these to flow cytometric analysis of major CD45+ cells collected including markers, at a minimum, for classical IEL populations and for eosinophils. Although not essential the tissues for this experiment would be available from the same animals as used for exp 2 above (day 14 cells samples would be sufficient) and the information would greatly strengthen the manuscript.

---

## [Author Response]

Essential revisions:Experiment 1) The reviewers agree that a more quantitative comparison of their flow results to histology would be very valuable to confirm how representative their flow results are in reflecting cell types present in the tissue. It would be sufficient to provide this for day 14 (using specific markers for eosinophils, CD4^+^ or CD4- lymphoid cells, B cells and macrophages). The reviewers acknowledge that no digestion protocol will be able to perfectly preserve cell ratios present in the tissue, but it would be valuable for the reader to have this information and to clearly state in the manuscript the caveats of using single cell analysis to describe cell ratios. Please note that quantitative graphs (cell counts or pixel area if positive staining per mm^2^ of tissue, using a set number of villi and tissue sections) should be provided.

We performed additional experiments to address this comment. As suggested by the reviewers, we stained duodenal sections from naïve and day 14 infected mice for B cells, eosinophils, macrophages and CD3+ CD4^+^ and CD3+ CD4- T cells and quantified cell counts per villi per mm^2^ and as a percentage of CD45+ cells. When we compared these frequencies to our spectral flow cytometry data we observed that the frequencies of B220+, CD3+ CD4^+^, and CD64+ cells were highly comparable between confocal microscopy and spectral flow cytometry, while Siglec F+ cells were overrepresented in our spectral flow cytometry data and CD3+ CD4- cells were underrepresented. However, changes within immune cell populations at the peak of *H. polygyrus* infection were faithfully reported by both confocal microscopy and spectral flow cytometry, reminding us that cell ratios defined by single cell analysis need to be carefully validated within the tissue before conclusions are drawn. We have incorporated these findings into the main text (Results and Discussion, paragraph eight) and presented the data in a new supplementary figure (Figure 2—figure supplement 5).

Experiment 2) To substantiate the claim that this techniques will be useful for single cell analysis it would be useful for the reader to know whether it can be employed for single cell RNA seq experiments. In the regard the authors should provide an indication of RNA quality able to be obtained from distinct cell populations harvested from the day 14 post-infection digest. Note that the reviewers acknowledge high quality RNA is difficult to obtain from neutrophils and eosinophils, thus it would be sufficient if the authors provided data for the other major cell subsets namely macrophages (CD64+MHCII+), T cells and B cells.

We performed additional experiments to address this comment. As suggested by the reviewers, we sorted 5,000 B cells, CD4 T cells and macrophages from naïve or day 14 *H. polygyrus* infected mice, extracted their RNA and assessed RNA quality using the Agilent TapeStation. Our RNA quality analysis resulted in high RIN numbers (range 6.8-10 for naïve and 7.0-10 for day 14 *H. polygyrus* samples). However, no clear separation of the 18S and 28S peaks could be observed on the gel image or the electropherograms (Figure 2—figure supplement 8C,D). While the RIN numbers might not have been correctly calculated, and a technical optimization of the TapeStation protocol might be necessary, no RNA degradation was observed in naïve and day 14 *H. polygyrus* samples, suggesting that the extraction of high-quality RNA is feasible from both naïve and day 14 *H. polygyrus* infected mice using our cell isolation protocol. We have incorporated these findings into the main text and presented the data in a new supplementary figure (Figure 2—figure supplement 8). We also got in touch with Agilent to get advice on how to improve the separation of the 18S and 28S peaks, but despite lengthy discussions and several trials no solution was immediately available.

Experiment 3) I would be very useful if the authors collected the cells released from the epithelial layer (intra-epithelial cells) during the EDTA and vortexing stages and subjected these to flow cytometric analysis of major CD45+ cells collected including markers, at a minimum, for classical IEL populations and for eosinophils. Although not essential the tissues for this experiment would be available from the same animals as used for exp 2 above (day 14 cells samples would be sufficient) and the information would greatly strengthen the manuscript.

We have repeatedly analysed the EDTA wash preparations and while we can easily detect CD45+ cells, eosinophils and many different populations of IELs from naïve animals, no CD45+ cell were detected in preparations from infected animals. We have mentioned this finding in the main text and clarified that our protocol should only be used to isolate lamina propria cells from infected intestines, as we believe that a completely new protocol needs to be optimised to obtain intra-epithelial cells from infected intestines, which is beyond the scope of this study.